# Periacetabular Tumour Resection under Anterosuperior Iliac Spine Allows Better Alloprosthetic Reconstruction than Above: Bone Contact Matters

**DOI:** 10.3390/jcm11154499

**Published:** 2022-08-02

**Authors:** Alessandro Bruschi, Luca Cevolani, Benedetta Spazzoli, Marco Focaccia, Stefano Pasini, Tommaso Frisoni, Davide Maria Donati

**Affiliations:** Unit of 3rd Orthopaedic and traumatologic Clinic Prevalently Oncologic, IRCCS Istituto Ortopedico Rizzoli, Via Pupilli 1, 40136 Bologna, Italy; luca.cevolani@ior.it (L.C.); benedetta.spazzoli@ior.it (B.S.); marco.focaccia@ior.it (M.F.); stefano.pasini@ior.it (S.P.); tommaso.frisoni@ior.it (T.F.); davidemaria.donati@ior.it (D.M.D.)

**Keywords:** pelvic resection, alloprosthetic composite, bone sparing, multiplanar osteotomies, surgical navigation, 3D custom made prosthesis

## Abstract

Background: Periacetabular resections are more affected by late complications than other pelvic resections. Reconstruction using bone allograft is considered a suitable solution. However, it is still not clear how the bone-allograft contact surface impacts on mechanical and functional outcome. Materials and methods: This paper presents the results of a retrospective analysis of 33 patients with resection of the entire acetabulum and reconstruction with an allograft-prosthetic composite for the period 1999 to 2010. Patients were divided in two groups, based on type of resection. In Group 1. patients had resections under anterosuperior iliac spine allowing the highest bone-allograft surface contact in reconstruction, while in Group 2 patients had resections over it. Results: Mechanical survival of the implant and Musculoskeletal Tumor Society functional score were calculated. Impact of age and artificial ligament were investigated as well. Patients in Group 1 had 38% mechanical failure rate of the implant while patients in Group 2 had 88%. Average functional score was higher in Group 1 compared with patients in Group 2. An artificial ligament was shown to have non-significant impact on survival of the reconstruction in Group 1, while significantly improving survival of reconstruction in Group 2. Conclusion: Bone-allograft contact matters: resection under anterosuperior iliac spine allows better mechanical survival and offers better reconstruction functional scores.

## 1. Introduction

Pelvic resections involving acetabulum are more affected by late complications than other pelvic resections, and, hence, the worst functional prognosis should be expected [1]. The resulting large bone and soft tissue defects, the perioperative complications and the post-operative tumour treatment frequently result in complications threatening the implant survival. Factors majorly determining complications are post-operative chemotherapy, the long surgical intraoperative time and the type of reconstruction [2,3,4,5,6]. Indeed, reaching a suitable reconstruction following this type of resection is a major challenge for the orthopaedic oncologist [3]. Different reconstructive techniques have been proposed over the years: Harrington reconstruction, iliofemoral coaptation or pseudoarthrosis, ischiofemoral arthrodesis, saddle prosthesis, alloprosthetics composite and 3D custom made prostheses [3,7,8]. When reconstruction is not a suitable option, flail hip can be performed [9]. Among all these techniques, alloprosthetics composite reconstruction consists of a bone allograft customized on the operative table to restore pelvic bone defect combined with a total hip arthroplasty. In most cases, it is a reliable solution after periacetabular tumour resection in terms of durability and function [3,6]. However mechanical, infective and local tumour relapse complications range from 30% to 90% [2,6,10,11,12]. Infection and dislocation are the most common complications in the early stages [2,3,6], while, loosening and mechanical failure are later complications [6,10]. However, type II pelvic resections can involve much more than the periacetabular area, and, thus, it is unclear if extension of the resection due to tumour location impacts on the functional outcome of the reconstruction over time [1]. The extension of resection is reported with no effect on outcome of a periacetabular reconstruction in [5,6], while other reports [5,11,13,14] showed better outcome in cases with limited bone excision. We, therefore, decided to retrospectively review our series of patients who experienced pelvic resection and reconstruction with allograft prosthetic composite to investigate if the extension of bone-allograft interface in the reconstruction impacts on survival of reconstruction, thus, pivoting the choice of the type of resection when this is possible.

## 2. Materials and Methods

### 2.1. Criteria of Data Collection

We retrospectively reviewed 69 patients treated with internal hemipelvectomy for a bone tumour involving the periacetabular area between November 1990 and August 2010. According to the Enneking and Dunham classification, all the patients had a type II alone or a combined type II resection; i.e., extending to ilium, ischium, pubis or sacrum, depending on tumour location [15]. Inclusion criteria: (a) patients with locally aggressive benign (Stage III) or malignant bone tumour of the periacetabular area; (b) diagnosis performed with either needle or incisional biopsy confirmed by an expert pathologist; (c) type II resection alone or combined type II resection of the pelvis; (d) follow up longer than 24 months; (e) no infection or local tumour relapse. Exclusion criteria: follow up less than 24 months; infective or local tumour relapse complications. According to the inclusion and exclusion criteria, 33 patients were considered for the retrospective analysis.

### 2.2. Classification System of Pelvic Resections Used

The 33 patients were divided into 2 groups following the classification system here suggested: Group 1 (resection under anterosuperior iliac spine allowing high bone-allograft contact in the reconstruction: 1A type resection) and Group 2 (resection above anterosuperior iliac spine allowing low bone-allograft contact in the reconstruction: all other resections) (Figure 1) (Table 1). 

The classification system was based on where the proximal osteotomy and the distal osteotomy were made at the time of surgery to achieve wide margins resection. We defined 4 regions in the super acetabular area (Figure 1) in which the proximal osteotomy had been performed in the different cases. Region 1 was contained between the roof of the acetabulum and a line that connects the apex of the greater sciatic notch and the anterosuperior iliac spine. Region 2 was comprised of the area between the anterosuperior iliac spine and the lateral margin of posterosuperior iliac spine. Region 3 was the area in between the lateral margin of the posterosuperior iliac spine bone and the sacroiliac joint. Region 4 included the sacroiliac joint, and thus, it was placed in the sacral wing. For the distal (medial) osteotomy, we identified three osteotomy locations: type A osteotomy involved the ipsilateral pubic ramus, type B osteotomy was in the pubic symphysis and type C involved the contralateral pubic ramus. 

The Senior Author (DD) participated in all procedures and more than 5 senior surgeons have been involved in the procedures during the considered period of time. In 4 patients only the extended iliofemoral (lateral) approach was used. In all other patients, inguinal extension was performed in order to better expose iliac vessels and nerves, including the obturator bundle. To improve the exposure of the posterior column of the acetabulum, in 7 patients a trochanteric osteotomy was performed and repaired using a cable grip device or a tension band wire. An extra-articular resection, removing the entire hip en bloc. during tumour excision was used in 9 patients. All procedures were performed using a frozen, non-irradiated, pelvic allograft shaped to match the bone excised. Grafts were thawed in Rifampin solution (Lepetit, Milan, Italy) for 60 min before use. Cultures for aerobic and anaerobic bacteria were performed after thawing. The first 9 patients received a reconstruction using a bipolar cup or a metal-backed cup cemented in the acetabular allograft. The other 14 patients were reconstructed with different acetabular cups. In all these cases, the acetabulum was cemented, and bone screws supplemented fixation of the cup to the allograft. To protect the allograft from possible fracture of the medial acetabular, a contoured neutralization plate along the innominate bone was used in 6 of these patients. The remaining 10 patients were treated with a McMinn acetabular prosthesis (Waldemar Link, Hamburg, Germany), which incorporated a large central stem, from 45 mm to 85 mm in length. We did not use additional screws for fixation, nor a contoured neutralization plate, in this type of reconstruction. In all procedures involving the iliac wing, we used three or four large cancellous screws fitted with washers (Synthes, Paoli, PA, USA) passing through the graft, into the sacrum (Figure 2). In allograft-host junctions at the symphysis pubis, an anterior plate was fixed onto the contralateral pubic bone with screws and locking nuts, or the symphysis was fixed with screws and cerclage wires. In 21 patients, to avoid early hip prosthetic dislocation, we decided to use one or two artificial ligaments. They were fixed to the proximal femur (inter trochanteric area) with a screw, after having passed the screw into the allograft ileopubic branch through drilled holes and twisting it around the prosthetic neck. A Ligament Advanced Repair System (LARS) artificial ligament was used in the reconstruction (Corin Group, Cirencester, UK) in Figure 2. 

### 2.3. Data Collection

According with the described classification, Group 1 included 16 patients who had received 1A resection, while Group 2 included 17 patients who had received other resections (two 1B, one 2A, two 2B, one 3B, seven 4A and four 4B). Characteristics of patients are recorded in Table 2. Resection margin results were wide in all patients.

In Group 1 (16 cases), six patients were younger than 40 years old at the time of the surgical procedure, while ten were older. Ten patients were treated with a hip artificial ligament.

In Group 2 (17 cases), eight patients were younger than 40 years old, while nine were older at the time of the surgical procedure. In 11 hip artificial ligaments were used in the reconstruction.

The end point of the analysis was composite revision, with either revision of the acetabular cup or allograft retrieval (including acetabular cup).

Patients had regular follow ups, every three months in the first three years, every six months in the fourth and fifth year and once a year thereafter. Functional status was calculated during follow-up through the Musculoskeletal Tumor Society evaluation score (MSTS93) [16]. 

All follow up radiographs were compared with previous ones: loosening was considered when the cup was displaced, compared to the previous follow-up. Reabsorption was defined when more than 1 cm of the allograft around the cup was eroded over time or when large progressive lucencies were seen around the fixation devices. Fracture of the allograft was considered when a bone interruption inside the body, or in the margins of the allograft, was seen. Dislocation of the hip prosthesis was considered when the femoral head lost articular contact with the acetabular cup. Breakage of plate and screws was evident when these fixation devices lost their integrity as one single element. Non-union was finally defined as failure of union by one year after surgery.

### 2.4. Statistical Analysis 

Statistical analysis was performed using the Statistical Package for the Social Sciences (SPSS) software version 21.0 (SPSS Inc., Chicago, IL, USA). Kaplan-Meier analysis with the log-rank test was used in order to calculate the failure rate and the mean failure time of the 2 groups [17]. In all reconstructions, the endpoint of the analysis was at least one surgical operation for revision of the implant. Functional evaluation of the patients in the two groups was recorded at two and five years, and at the final follow up. The planned functional evaluation was completed in all but eight patients. Two of them lacked the two years follow up, five the five years follow up and one lacked the final follow up. Cox regression multivariate analysis (with Wald’s backward method) was used to calculate the impact of age (</>40 years old) and artificial ligament on failure rate in the two groups at 120 months (Table 2).

## 3. Results

### 3.1. Impact of Bone-Allograft Contact on the Reconstruction

Patients in Group 1 had a reconstruction with better survival and functional scores compared to patients in Group 2. In Group 1 the implant had a 38% failure rate and a mean failure time of 144 months (+/−69 months). While patients in Group 2 had an 88% failure rate with a mean failure time of 109 months (+/−84 months) (95% CI, *p* = 0.021) (Figure 3). The endpoint of analysis was at least one surgical operation for revision of implant due to mechanical failure (aseptic loosening, fracture or resorption of the allograft, dislocation or breakage of the hardware).

Post-hoc sample size analysis for survival of implant (16 patients in Group 1 and 17 patients in Group 2, with 38% failure rate in Group 1 and 88% in Group 2, *p* > 0.05) showed power of the study to be 88%. 

In Group 1, average functional score was 81% at two years follow up, 77% at five years and 68% at final follow up (mean time 145 months). In Group 2 the average functional score at two, five years and at the final follow up (mean time 109 months) were 62%, 70% and 54%, respectively. 

In Group 1 (16 patients), failure occurred in 7 patients (%). Six of these patients presented implant failure for loosening of the acetabular cup and were treated with acetabular cup substitution. Another patient failed due to reabsorption of the acetabular wall and was treated with substitution of the acetabular cup with a cemented one. None of these patients underwent allograft removal. Of the six failed patients, in three of them an artificial ligament had been used in the primary reconstruction.

In Group 2 (17 patients), failure occurred in 15 patients (%). Seven patients presented cup loosening and were treated with acetabular cup revision (one 1B, one 2B, one 3B, three 4B and one 4B). Four patients had a fracture of the allograft, treated with acetabular cup revision with an acetabular reinforcement cage in one case (2A), removal of the composite and saddle prosthesis in another case (4B), removal of the implant and flail hip in another patient (1B) and acetabular cup revision with iliac crest bone autograft in the other one (4A). One patient underwent a femoral head revision for dislocation (4A), one patient had a revision of the implant without allograft removal, due to breakage of the neutralization plate (4B). Finally, two more patients presented with non-union of the bone graft. One had the implant and the graft removed and substituted by saddle prosthesis (2B); while, in the other. the implant was revised with an iliac bone crest autograft without allograft removal (4A). In six failures of Group 2 an artificial ligament had not been used, three of them failed with removal of the allograft.

### 3.2. Impact of Age on the Reconstruction

In both groups, patients older than 40 years old had lower failure rates. In Group 1, patients older than 40 years old had a survival rate of 90%, while patients younger than 40 years old had 67% survival at 120 months (CI 95%, *p* = 0.048). In Group 2, patients older than 40 years old had a survival rate of 56%, while patients younger than 40 years old had a 38% survival rate at 120 months (CI 95%, *p* = 0.048).

### 3.3. Impact of Artificial Ligament on the Reconstruction

In Group 1 there was no significant difference in implant survival if the artificial ligament was used or not. On the other hand, in Group 2, survival of the reconstruction benefited from the use of the artificial ligament.

Considering patients in Group 1, those in whom an artificial ligament was used in the reconstruction had an 83% survival rate, while those with no artificial ligament used in the reconstruction had an 80% survival rate, at 120 months (CI 95%, *p* = 0.02) (Figure 4).

In Group 2, those in whom an artificial ligament was used in the reconstruction had a 64% survival rate, while those with no artificial ligament used in the reconstruction had a 17% survival rate, at 120 months (CI 95%, *p* = 0.02) (Figure 5).

## 4. Discussion

This study was a retrospective analysis on a limited number of patients, but due to rarity of the site of operation and the need for long term follow-ups this case series was larger than others in the literature. 

### 4.1. Impact of Bone-Allograft Contact on the Reconstruction

Reconstruction in Group 1 had a better implant survival if compared to Group 2, scoring a failure rate of 38% vs. 88% (average 144 months +/−69 months vs. 109 +/−84). Concerning functional score, Group 1 scored 81%, 77% and 68% vs. 62%, 70% and 54%, respectively, at two year follow up, five year follow up and final follow up. These results indicate that providing higher bone-allograft contact allows more mechanical durability and higher performance of functional reconstruction. These results are comparable with those of Beadel et al., where type II pelvic resections had a better functional score than type II combined [18]. In our opinion, our results integrate Beadel’s conclusion under a mechanical point of view, as in that series deep infection severely impaired the mechanical outcome of larger resections [18]. Other series, considering resection type, also suggest that larger resections are consistently influenced by related deep infection, so, the potential mechanical outcome remains unclear in those series as they appear to be indirectly influenced by infection [4,5,19]. Compared with the MSTS reported in literature (average functional result 70%), in our study Group 1 scored higher and Group 2 lower than 70% [3,4,6]. One of the reasons to explain the better results in Group 1 is related to the direction of the iliac osteotomy. In Group 1, the osteotomy under the anterosuperior iliac spine endures less vertical mechanical shear stress at the bone-allograft interface (Figure 6); therefore, multiplanar osteotomies could be suitable options to tackle this problem in proximal osteotomies over to the anterosuperior iliac spine [20] (Figure 7). On the other hand, the distal osteotomy (pubic area) affects pelvic ring integrity; in type A resections pubic symphysis is spared, while it is not spared in types B and C [21,22]. This factor could contribute to better results in 1A resections. 

Long-term survival and functional results of alloprosthetic composite reconstruction in this series was fair. However, we believe surgically navigated resections and 3D custom made reconstructions assisted by patient specific instrumentation should provide better results [20,23,24,25,26,27]. Actually, some studies highlight the importance of bone stock sparing resections in order to improve mechanical properties of the reconstruction, such as in the following scenarios, which preserve wide oncological margins while sparing the bone mass needed for a suitable bone-reconstruction contact: iliac multiplanar osteotomies, surgically navigated resections and 3D custom made reconstructions [13,14,20,23,24,28].

### 4.2. Impact of Age on the Reconstruction

In both groups, patients older than 40 years old had lower failure rates at 120 months. The lifestyle and the higher functional requests of a younger patient could justify mechanical overuse and earlier failure of the implant. Influence of age on cementless fixation is of paramount interest in orthopaedics, as most orthopaedic implants are placed in elderly bone tissues. Despite the fact that it is reported that bone and mineral metabolism decreases with ageing, the impact of age on bone-implant interface remains unclear in humans [29]. Almost all studies investigating this topic have been conducted on animals presenting contrasting results: some studies state ageing decreases bone callus formation and causes lower bone ingrowth in porous implants [29], while in other studies older animals present lower interface shear stress and comparable fixation survival if compared to younger animals [18,22,30,31]. Despite ageing probably having a negative effect on bone-allograft union, these works support the idea that the higher shear stress forces in younger patients can overcome the good performance of a younger bone.

### 4.3. Impact of Artificial Ligament on the Reconstruction

In Group 1, the use of an artificial ligament to stabilize hip articulation had no significant impact on the mechanics of the reconstruction. On the other hand, in Group 2, the artificial ligament significantly impacted on the mechanics of reconstruction, giving 64% and 17% survival rates, at 120 months. The mechanical role of artificial ligaments in improving mechanics of the reconstruction is well known [6,32,33]. However, due to lower soft tissue excision, a limited resection probably has lower impact on hip stability. The correct indication for the use of artificial ligament is to be defined, as it has been demonstrated to increase the risk of infection and synovitis in some series [34,35,36]. Therefore, in reconstruction with limited soft tissue sacrifice, this risk could be avoided without any consequence on the stability of the reconstruction. 

## 5. Conclusions

In conclusion, where periacetabular tumour resection is performed can influence reconstruction survival and function. In this study, resections under anterosuperior iliac spine allowed more bone-allograft contact in the reconstruction and were demonstrated to perform better in terms of long-term failure (38% at long term follow up), yielding better functional outcomes (75%), if compared to resections above the anterosuperior iliac spine. Studies comparing functional results after resections inside the iliac wing (low contact reconstruction) and after resections directly in the sacroiliac joint (higher contact reconstruction) should be considered. These studies could demonstrate that pursuing a higher contact reconstruction may provide better functional results, even at the cost of a wider resection. Furthermore, older age and the use of an artificial ligament in selected cases showed better clinical and functional scores. In the future, multiplanar osteotomies, surgical navigation and 3D custom made prostheses could improve the results of all resections, allowing better bone-reconstruction contact.

## Figures and Tables

**Figure 1 jcm-11-04499-f001:**
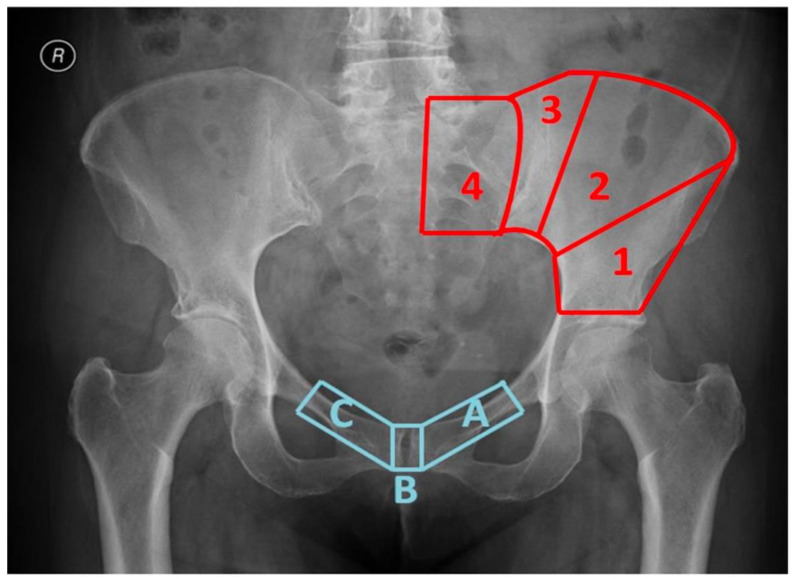
Classification system based on the amount of bone resected. Numbers are related to the four different regions in which the proximal osteotomy (super acetabular) can be made. Letters define the distal osteotomy (medial) location.

**Figure 2 jcm-11-04499-f002:**
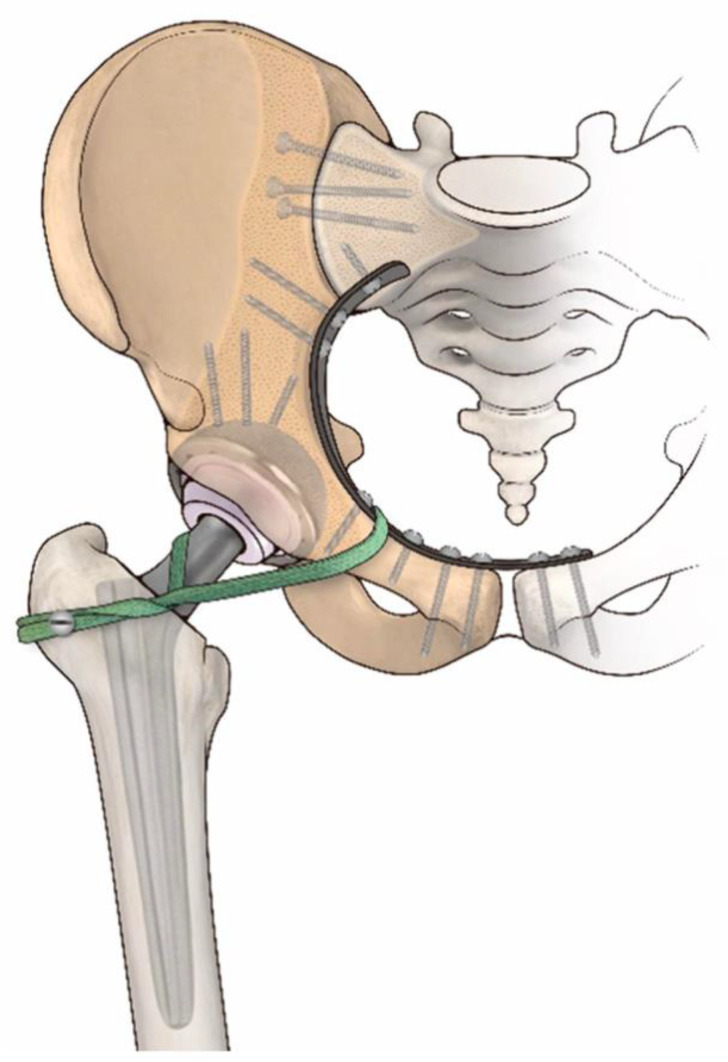
Ileo-femoral artificial ligament used for immediate stabilization of the hip joint.

**Figure 3 jcm-11-04499-f003:**
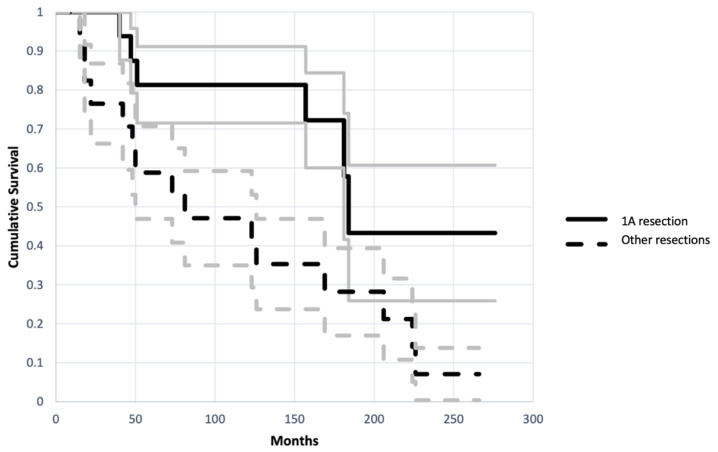
Kaplan-Meier plot shows different cumulative survival in Group 1 (A1 resection) and in Group 2 (Other resections) (CI 95%, *p* = 0.02). Endpoint of analysis was at least one surgical operation for revision of implant due to mechanical failure (aseptic loosening, fracture or resorption of the allograft, dislocation or breakage of the hardware). Grey lines represent standard deviation.

**Figure 4 jcm-11-04499-f004:**
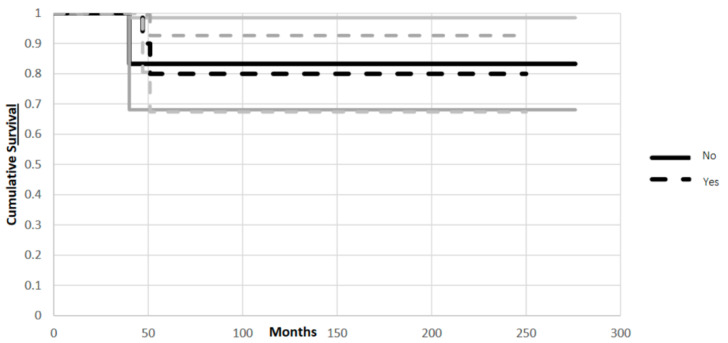
Kaplan-Meier plot shows similar survival rate in Group 1 with ligament in the reconstruction (Yes) or without artificial ligament in the reconstruction (No), at 120 months (CI 95%, *p* = 0.02). The endpoint of analysis was at least one surgical operation for revision of implant due to mechanical failure (aseptic loosening, fracture or resorption of the allograft, dislocation or breakage of the hardware). Grey lines represent standard deviation.

**Figure 5 jcm-11-04499-f005:**
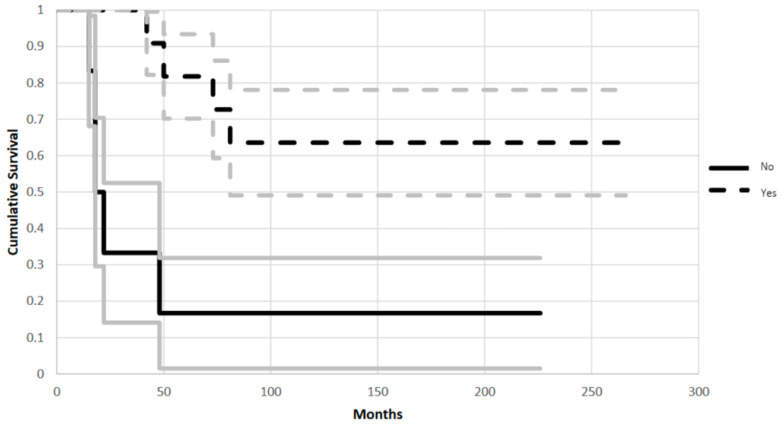
Kaplan-Meier plot shows improved mechanical survival in Group 2 (Yes) using artificial ligament, at 120 months (CI 95%, *p* = 0.02). The endpoint of analysis was at least one surgical operation for revision of implant due to mechanical failure (aseptic loosening, fracture or resorption of the allograft, dislocation or breakage of the hardware). Grey lines represent standard deviation.

**Figure 6 jcm-11-04499-f006:**
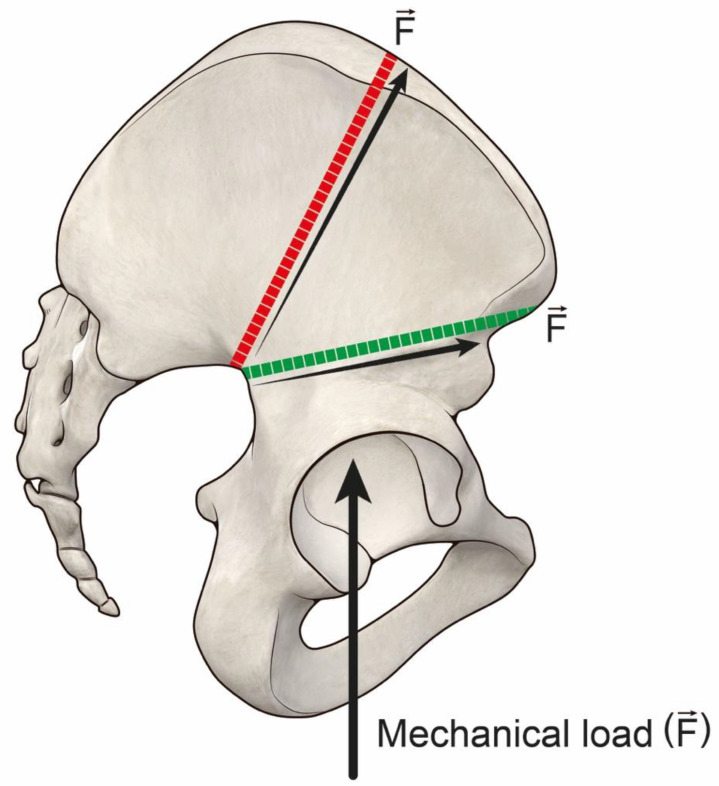
Shear stress vertical forces are less when the osteotomy line is made under the anterosuperior iliac spine.

**Figure 7 jcm-11-04499-f007:**
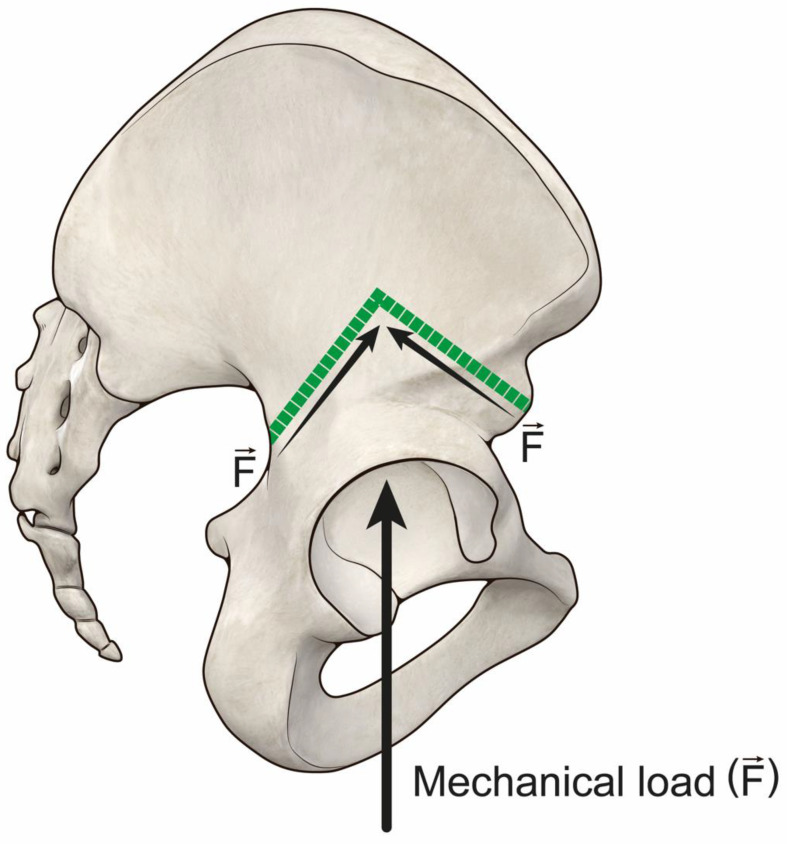
Biomechanics of a multiplanar osteotomy: the mechanical load is distributed along the two osteotomy lines with limited vertical shear stress.

**Table 1 jcm-11-04499-t001:** Classification System.

Region of Proximal Osteotomy Line:	Region of Distal Osteotomy Line:
**1**Osteotomy inside an area comprised between the roof of the acetabulum and a line that connects the apex of the ischiatic notch and the anterosuperior iliac spine	**A**Osteotomy involving the ipsilateral pubic ramus
**2**Osteotomy inside an area comprised between the anterosuperior iliac spine and the lateral margin of posterosuperior iliac spine bone bulk	**B**Osteotomy involving the pubic symphysis
**3**Osteotomy inside an area comprised between the lateral margin of posterosuperior iliac spine bone bulk and the sacro-iliac joint (excluded)	**C**Osteotomy involving the contralateral pubic ramus
**4**Osteotomy medial to the sacro-iliac joint (included)	

**Table 2 jcm-11-04499-t002:** Data on the 33 patients of the study. Chs c = central chondrosarcoma; Chs dediff = dedifferentiated chondrosarcoma; GCT = giant cells tumour; Os L = low grade osteosarcoma; Os H = high grade osteosarcoma; Spindle cell s = spindle cell sarcoma; Ewimg = Ewing sarcoma; Angios = angiosarcoma; Dod = dead of disease; Ned = no evidence of disease.

Number	Gender	Age	Diagnosis	Stage	Resection Type	Postoperative Chemotherapy	Patient Follow Up (Months)	Status
1	M	40	Chs c	II B	1A	No	194	Ned
2	M	32	Chs c	I B	1A	No	187	Ned
3	F	56	Chs c	I A	2A	No	256	Ned
4	F	69	Chs c	I B	1A	No	168	Ned
5	M	34	GCT	3	1A	No	286	Ned
6	F	59	Chs c	I B	4A	No	255	Ned
7	M	38	Chs c	I A	1A	No	194	Dod
8	M	62	Chs c	IB	1A	No	110	Ned
9	F	49	Chs dediff	II B	4B	Yes	100	Dod
10	F	25	Chs c	I B	3B	No	277	Ned
11	M	22	Os L	I A	2B	No	258	Ned
12	F	17	Os H	II B	4A	Yes	252	Ned
13	F	22	Os H	II B	4A	Yes	252	Ned
14	M	21	Chs c	IIB	4A	No	295	Ned
15	F	33	GCT	3	1A	No	230	Ned
16	M	57	Chs dediff	II B	4B	No	300	Ned
17	M	56	Chs c	IB	1A	No	142	Ned
18	M	60	Chs c	I B	4B	No	146	Ned
19	M	65	Chs c	II B	2B	No	266	Dod
20	F	37	Chs c	II B	4B	No	305	Ned
21	M	51	Chs c	I B	1B	No	138	Ned
22	F	63	Angios	IIB	4A	No	38	Dod
23	M	29	Tcg	III	4A	No	331	Ned
24	F	51	Os H	II B	1A	Yes	65	Dod
25	M	48	Chs c	I B	1A	No	169	Ned
26	M	66	Chs c	I B	1A	No	123	Ned
27	F	24	Chs dediff	II B	1A	No	276	Ned
28	F	43	Os H	I B	1A	No	167	Ned
29	M	59	GCT	3	1A	No	250	Ned
30	F	35	Spindle cell s	II B	4A	Yes	184	Ned
31	M	53	Chs c	I A	1B	No	173	Ned
21	F	59	Chs c	I B	1A	No	51	Ned
33	M	18	Ewing	II	1A	Yes	157	Ned

## Data Availability

Not applicable.

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
