# Peer review of "Periacetabular Tumour Resection under Anterosuperior Iliac Spine Allows Better Alloprosthetic Reconstruction than Above: Bone Contact Matters"

_jcm, 2022, doi:10.3390/jcm11154499_

Round 1

Reviewer 1 Report

This is a paper regarding the survivorships of reconstruction surgery performed combined with tumor resection. Authors declare that better a survival rate would be provided if the osteotomy was performed at a level lower than ASIS.

The conclusion sounds reasonable, but the data presented was too simplified as a scientific article. Authors should define and count the mode of failure (dislocation, loosening, migration of implant, infection and breakage of the implant), and then attempt to relate those complications to the area of resection.

The sites of resection were divided into 2 groups in the text, but into 4 groups in the Table1 and Figure 1.

Figure 3

What was the endpoint for those survival rates? Death? Implant revision? Any additional surgery?

Endpoint was mentioned in Statistical analysis, but the endpoint should be defined each time because sometimes researchers define several different endpoints in single paper. Without clarification, readers need to go back and read method section each time.

A1 resection? 1A resection? Authors should try their best not to confuse readers. They mentioned a lot of classifications and readers would stop reading unless authors use those term clearly and accurately.

Figure 4

What was the endpoint?  Dislocation? Additional surgery?

Figure 5

Please state clearly what was the mechanical survival. The endpoint is unclear. Readers imagine that would be dislocation or aseptic loosening, but nobody would cite the paper if the endpoint was not defined.

Please describe surgical procedure for each resection type. How to fix the implant, name of implant, and how to use of the artificial ligament.

Author Response

We thank the reviewer for pointing very important issues that otherwise would have affect the quality of the paper.

Point 1: The conclusion sounds reasonable, but the data presented was too simplified as a scientific article. Authors should define and count the mode of failure (dislocation, loosening, migration of implant, infection and breakage of the implant), and then attempt to relate those complications to the area of resection.

Reply to point 1: Amended in the text in red from line 195 to line 213.

Point 2: The sites of resection were divided into 2 groups in the text, but into 4 groups in the Table1 and Figure 1.

Reply to point 2: This is because Figure 1 and Table 1 explain the classification system. Those in red aren't for groups, but those are the four zones in which is possible to perform the proximal osteotomy (than A,B,C in blu are the three zones for the distal osteotomy). On the other side, we decided to divide the patients into two groups for the analysis, due to the number of patients and to the main aim to use the ASIS as marker point for predicting different functional outcomes.

Point 3: Figure 3

What was the endpoint for those survival rates? Death? Implant revision? Any additional surgery?

Endpoint was mentioned in Statistical analysis, but the endpoint should be defined each time because sometimes researchers define several different endpoints in single paper. Without clarification, readers need to go back and read method section each time.

Reply to point 3: Amended in the text in red in lines 184-186, 236-238, 243-245 and 249-251. The endpoint of analysis is at least one surgical operation for revision of implant, due to mechanical failure (aseptic loosening, fracture or resorption of the allograft, dislocation or breakage of the hardware).

Point 4: A1 resection? 1A resection? Authors should try their best not to confuse readers. They mentioned a lot of classifications and readers would stop reading unless authors use those term clearly and accurately.

Reply to point 4: amended in Figure 3. The number is put first (proximal osteotomy) and the letter for second (distal osteotomy).

Point 5: Figure 4

What was the endpoint?  Dislocation? Additional surgery?

Reply to point 5: Amended in the text in red in lines 243-245

Point 6: Figure 5: Please state clearly what was the mechanical survival. The endpoint is unclear. Readers imagine that would be dislocation or aseptic loosening, but nobody would cite the paper if the endpoint was not defined.

Reply to point 6: Amended in the text in red lines 249-251.

Point 7: Please describe surgical procedure for each resection type. How to fix the implant, name of implant, and how to use of artificial ligament.

Reply to point 7: The text has been integrated with description of surgical procedures, name of implants and surgical technique for artificial ligament in red from line 105 to 133.

Reviewer 2 Report

The authors presented a comparison of two methods in reconstruction after periacetabular resections. This is a retrospective study with limited sample size. Considering that these kind of cases are rare, this research is still of value. To guarantee the quality of this paper, I suggest the researchers to do more analysis.

1. Please add a post-hoc sample size analysis to determine the robustness of this comparison.

2. Since the period is long, which may introduce bias of learning-curve, please mention the number of senior surgeons involved in the procedures, 

3. Please directly add a table of basic characteristics comparison between two groups, so as to eliminate confounders. If there are factors significantly different between two groups, correlation/regression analysis is necessary.

Author Response

We thank the reviewer for pointing very important issues that otherwise would have affect the quality of the paper.

Point 1: Please add a post-hoc sample size analysis to determine the robustness of this comparison.

Reply to point 1: Post-hoc sample size analysis result stated in green in lines 187-189.

Point 2: Since the period is long, which may introduce bias of learning-curve, please mention the number of senior surgeons involved in the procedures.

Reply to point 2: Integrated the text in green lines 104-105.

Point 3: Please directly add a table of basic characteristics comparison between two groups, so as to eliminate confounders. If there are factors significantly different between two groups, correlation/regression analysis is necessary. 

Reply to point 3: We included all basics characteristics and type of resection for each patient in Table 2. In our opinion the characteristics of patients do not significantly differ between 1A resection patients and patients who received other resections considered all together.

Round 2

Reviewer 1 Report

All issues I raised have been addressed appropriately.  I think authors made good efforts to improve the manuscript.